# Genetically Modified Mesenchymal Stromal/Stem Cells as a Delivery Platform for SE-33, a Cathelicidin LL-37 Analogue: Preclinical Pharmacokinetics and Tissue Distribution in C57BL/6 Mice

**DOI:** 10.3390/antibiotics14050429

**Published:** 2025-04-24

**Authors:** Vagif Ali oglu Gasanov, Dmitry Alexandrovich Kashirskikh, Victoria Alexandrovna Khotina, Arthur Anatolievich Lee, Sofya Yurievna Nikitochkina, Daria Mikhailovna Kuzmina, Irina Vasilievna Mukhina, Ekaterina Andreevna Vorotelyak, Andrey Valentinovich Vasiliev

**Affiliations:** 1Koltzov Institute of Developmental Biology of Russian Academy of Sciences, Moscow 119334, Russia; 2Department of Normal Physiology, Privolzhsky Research Medical University of Ministry of Health of the Russian Federation, Nizhny Novgorod 603005, Russia; 3Department of Cell Biology, Biological Faculty, Lomonosov Moscow State University, Moscow 119234, Russia

**Keywords:** mesenchymal stromal/stem cells, antimicrobial peptides, cathelicidins, pharmacokinetics, biodistribution, cell-based therapy

## Abstract

Background: The genetic modification of mesenchymal stromal/stem cells (MSCs) to express antimicrobial peptides may provide a promising strategy for developing advanced cell-based therapies for bacterial infections, including those caused or complicated by antibiotic-resistant bacteria. We have previously demonstrated that genetically modified Wharton’s jelly-derived MSCs expressing an antimicrobial peptide SE-33 (WJ-MSC-SE33) effectively reduce bacterial load, inflammation, and mortality in a mouse model of *Staphylococcus aureus*-induced pneumonia compared with native WJ-MSCs. The present study aimed to evaluate the pharmacokinetics and tissue distribution of the SE-33 peptide expressed by WJ-MSC-SE33 following administration to animals. Methods: WJ-MSC-SE33 were administered to C57BL/6 mice at therapeutic and excess doses. The biodistribution and pharmacokinetics of the SE-33 peptide were analyzed in serum, lungs, liver, and spleen using chromatographic methods after single and repeated administrations. Results: The SE-33 peptide exhibited dose-dependent pharmacokinetics. The highest levels of SE-33 peptide were detected in the liver and lungs, with persistence in tissues for up to 48 h at medium and high doses of administered WJ-MSC-SE33. A repeated administration of WJ-MSC-SE33 increased SE-33 levels in target organs. Conclusions: The SE-33 peptide expressed by genetically modified WJ-MSCs demonstrated predictable pharmacokinetics and effective biodistribution. These findings, together with the previously established safety profile of WJ-MSC-SE33, support its potential as a promising cell-based therapy for bacterial infections, particularly those associated with antibiotic resistance.

## 1. Introduction

The antimicrobial resistance of pathogenic bacteria, which causes a wide range of infectious diseases, poses a serious threat to global public health. This challenge is closely associated with high morbidity and mortality rates, ineffective treatments, and increased healthcare costs worldwide [1,2]. A recent meta-analysis of clinical data from 2019 estimated that bacterial infections resistant to antimicrobials directly contributed to 3.62–6.57 million deaths, with 1.27 million attributed specifically to bacterial antibiotic resistance [1,3]. Particularly concerning is the rapid emergence of resistance in bacteria such as *Staphylococcus aureus*, *Klebsiella pneumoniae*, *Pseudomonas aeruginosa*, and *Streptococcus pneumoniae*, which cause infectious respiratory diseases [4,5,6]. For example, metatranscriptomic analyses have revealed that the expression of resistance genes to beta-lactams, aminoglycosides tetracyclines, and other antimicrobial agents can increase over a lifetime [7].

The problem of antimicrobial resistance is expected to aggravate the healthcare crisis in the near future. Pathogenic bacteria are able to survive antibiotic therapy by acquiring resistance genes through long-term selection processes [4,8]. These genes become the basis for the development of bacterial survival mechanisms, including bacterial colonization, biofilm formation, metabolic adaptations, drug-neutralizing pathways, and the efflux pump mechanism that removes components of antibacterial drugs from cells [9,10,11]. In this regard, novel therapeutic strategies are urgently required, such as cell-based therapies and antimicrobial peptides (AMPs) [12,13].

In recent years, mesenchymal stromal/stem cells (MSCs), including genetically modified MSCs, have garnered considerable interest as a potential therapeutic option for various diseases, including pulmonary infections [14,15,16,17,18,19]. The therapeutic utility of MSCs lies in their distinctive characteristics and properties, such as immunomodulatory, anti-inflammatory effects, anti-apoptotic activity, tissue regeneration, and antimicrobial effects [20,21,22]. Accordingly, MSCs play a critical role in the regeneration of tissues damaged by infectious agents, modulating immune cell infiltration, accumulation, and activity (e.g., neutrophils, monocytes, and macrophages) at sites of injury and infection. It is assumed that the activity of MSCs involved in bacterial clearance may be associated with direct intercellular interaction between MSCs and pathogens [23]. A significant role in the interaction between MSCs and pathogenic bacteria is attributed to their cellular secretome, which includes microvesicles and exosomes [24,25,26].

Indeed, the MSC secretome has been shown to exhibit antimicrobial effects against pathogens such as *Escherichia coli*, *S. aureus*, *S. pneumonia*, and *P. aeruginosa* [27,28,29,30]. For example, supernatants from MSCs have been shown to enhance the phagocytosis of *E. coli* by granulocytes and induce neutrophil chemotaxis [31]. This antimicrobial activity may, in part, be mediated by the secretion of AMPs such as LL-37 (hCAP-18), a human cathelicidin. This peptide inhibits bacterial growth through lysis and increasing their sensitivity to antibiotics [32,33,34]. LL-37 exerts broad-spectrum activity against Gram-positive and Gram-negative bacteria, as well as antiviral, antiparasitic (including antimalarial), antifungal, and antitumor properties. Additionally, LL-37 plays a role in modulating immune responses and inhibiting the activity of specific enzymes [35,36,37,38,39]. It is the only antibacterial peptide from the cathelicidin subfamily synthesized in the human body [36]. In addition to MSC, other cell types such as neutrophils, monocytes, macrophages, mast cells, NK cells, T and B lymphocytes, mucosal epithelial cells, adipocytes, and keratinocytes are also capable of producing LL-37 [32,40,41,42]. For instance, an increase in LL-37 production by bone marrow-derived MSCs was found in response to interaction with *E. coli*, and the antimicrobial effect has been observed in response to interaction with *E. coli*, with antimicrobial effects demonstrated both in co-culture systems of MSCs and *E. coli* and in an in vivo mouse model of pneumonia [32]. Similarly, another study revealed that microvesicles isolated from the conditioned media of MSCs increased LL-37 concentrations in the alveoli and plasma of rats [24].

LL-37 has been shown to alleviate pneumonia caused by various infectious agents, including *P. aeruginosa* and *S. aureus*, by mitigating the inflammatory response [43,44]. Recent studies using transgenic mice expressing the human *hCAP-18* gene demonstrated that LL-37 protects the lungs from bacterial infection and associated inflammation by significantly enhancing bacterial clearance and reducing the expression of key proinflammatory cytokines, such as IL-6 and IL-1β [44]. A similar effect was observed in an in vivo model of bacterial lung infection or acute lung inflammation, where genetically modified distal airway stem/progenitor cells expressing LL-37 were transplanted into mice. LL-37 expression in these cells not only protected mice from pneumonia and hypoxemia by enhancing bacterial clearance but also promoted the regeneration of damaged lung tissue [44]. Thus, genetically modified MSCs expressing AMPs such as LL-37 represent a promising therapeutic approach for the treatment of bacterial pneumonia.

However, it should be noted that the application of the previously mentioned LL-37 is associated with certain limitations [45,46,47,48]. Due to its susceptibility to proteolytic degradation, the resulting fragments of LL-37, as well as LL-37 itself in high concentrations, can exhibit a significant toxicity toward human cells and contribute to the development of additional chronic inflammatory conditions [48,49]. Furthermore, in recent years, several pathogens have been reported to develop adaptive mechanisms to LL-37, which attenuate its antimicrobial activity [46,47]. In this regard, the development of new antibacterial peptides that demonstrate a high efficacy by retaining the beneficial properties of natural analogues while eliminating undesirable effects represents a promising direction in the search for novel therapeutic approaches [50]. Currently, a wide array of peptides that are derivatives of LL-37 or its analogues has been developed [48,50,51,52]. Notably, a novel peptide, SE-33, has recently been synthesized, which is a truncated retro-analogue of LL-37 and exhibits a significantly reduced toxicity [53,54,55]. Structurally, the SE-33 peptide represents the retrosequence of the C-terminal region of the LL-37. It has been confirmed that this synthetic peptide possesses both antibacterial and antifungal activity [53,54,55].

Recently, we have demonstrated that genetically modified Wharton’s Jelly-derived MSCs (WJ-MSCs) engineered to express the antimicrobial peptide SE-33 exhibit a safety profile comparable to native MSCs, with no evidence of acute or chronic toxicity, dysfunction, or pathological alterations upon single or repeated administrations to animals [56]. Moreover, we have demonstrated that WJ-MSCs genetically modified to express the SE-33 peptide exhibit significant antimicrobial activity against *S. aureus*-induced bacterial pneumonia, leading to a reduction in mortality and enhanced bacterial clearance from the lungs compared to native WJ-MSCs. Furthermore, the observed decrease in leukocyte infiltration and inflammatory responses in lung tissue suggests that SE-33 may possess immunomodulatory properties that augment the intrinsic antimicrobial effects of MSCs. Thus, the genetic modification of MSCs to enable the secretion of the antimicrobial peptide SE-33 represents a promising approach for developing advanced cell-based therapies of pulmonary diseases, particularly those caused or complicated by bacterial infections.

Given the significant therapeutic potential of MSCs, understanding their systemic distribution and biodistribution is essential for assessing both their safety and therapeutic efficacy. However, the in vivo pharmacokinetics and biodistribution of the SE-33 peptide expressed by genetically modified WJ-MSCs remain unknown. Therefore, the aim of this study was to investigate the pharmacokinetics, tissue distribution, and systemic retention of the SE-33 peptide following both single and repeated intravenous administration to C57BL/6 mice.

## 2. Results

### 2.1. Pharmacokinetics of SE-33 Peptide in Mice Following the Single and Repeated Intravenous Administration of WJ-MSC-SE33

The pharmacokinetic profile of the antimicrobial peptide SE-33, expressed by genetically modified WJ-MSCs (WJ-MSC-SE33), was evaluated in C57BL/6 mice following a single intravenous administration of WJ-MSC-SE33 at three doses (0.5 × 10^7^, 1.25 × 10^7^, and 2.5 × 10^7^ cells/kg). SE-33 peptide levels were measured in the serum and peripheral organs, including lungs, liver, and spleen.

No detectable levels of SE-33 were observed in control animals receiving 0.9% NaCl or native WJ-MSCs, as indicated by the absence of characteristic chromatographic peaks in the 9.10–9.20 min retention time range on HPLC analysis (Appendix A). Furthermore, no detectable SE-33 peptide was observed in either serum or tissue samples collected at the 0 h pre-dose time point, confirming the absence of endogenous peptide expression prior to WJ-MSC-SE33 administration.

The lungs exhibited a dose-dependent increase in SE-33 concentration, with maximum concentrations (C_max_) of 0.233, 0.320, and 0.352 μg/mL corresponding to 0.5 × 10^7^, 1.25 × 10^7^, and 2.5 × 10^7^ cells/kg, respectively (Figure 1A). The peptide remained detectable in lung tissue for up to 20 h at the lowest dose and up to 48 h at higher doses. The area under the concentration–time curve from zero to infinity (AUC_0–∞_) values increased proportionally with dose (3.18, 6.50, and 9.37 h×μg/mL), reflecting 2.04- and 2.95-fold increases relative to the lowest dose (Table 1). Correspondingly, the elimination half-life time (T_1/2_) ranged from 12.07 h at the lowest dose to 16.00 h at the highest dose. The elimination half-life time (T_1/2_) extended with increasing doses (4.34, 14.3, and 22.9 h), alongside elevated AUMC_0–∞_ (32.9 to 292 h^2^×μg/mL) and mean residence time (MRT_0–∞_) values (10.4 to 31.2 h), indicating a dose-dependent prolongation of pulmonary exposure and tissue retention.

The liver exhibited the highest SE-33 concentrations among the examined tissues. C_max_ values reached 0.965, 1.235, and 1.428 μg/mL at increasing doses (Figure 1B). The peptide was detectable for up to 48 h across all doses of administered WJ-MSC-SE33. AUC_0–∞_ values increased from 24.0 to 38.0 h×μg/kg, with corresponding AUMC_0–∞_ values of 489, 702, and 833 h^2^×μg/mL (Table 1). However, unlike the lungs, hepatic T_1/2_ remained relatively stable (11.4–12.4 h) across doses, suggesting a saturation of clearance mechanisms or inherent metabolic stability. MRT_0–∞_ values ranged between 20.4 h and 21.9 h, supporting a consistent hepatic retention and potential metabolic processing of the peptide.

In serum samples, SE-33 was detected at lower concentrations relative to the liver but demonstrated a clear dose-dependent increase: 0.142, 0.223, and 0.262 μg/mL at the respective doses (Figure 1C). The peptide was detectable for 20, 24, and 36 h, respectively. AUC_0–∞_ values followed the same dose-dependent pattern (2.04, 3.73, and 5.17 h×μg/mL), while AUMC_0–∞_ increased from 21.2 to 79.7 h^2^×μg/mL, and MRT_0–∞_ increased from 10.4 to 15.4 h (Table 1). Notably, serum T_1/2_ values remained relatively consistent (4.80 to 4.00 h), and the elimination rate constant (λ_z_ = 0.144–0.173 h^−1^) indicated that the systemic clearance of SE-33 occurred independently of the administered dose.

The spleen demonstrated the lowest levels of SE-33 among all the evaluated tissues. At 0.5 × 10^7^ cells/kg, the peptide remained below the detectable threshold, reaching 0.043 μg/mL at 8 h. At higher doses, C_max_ values were 0.072 μg/mL and 0.133 μg/mL (Figure 1D). The peptide was detectable for up to 20 h, while AUC_0–∞_ values reached only 1.03 and 1.81 h×μg/mL at 1.25 × 10^7^ and 2.5 × 10^7^ cells/kg, respectively. Splenic AUMC_0–∞_ (11.6 and 18.1 h^2^×μg/mL), MRT_0–∞_ (11.3 and 10.0 h), and T_1/2_ (5.09–5.59 h) were significantly shorter than those in other tissues, indicating a rapid clearance from this compartment, likely reflecting active immune-mediated clearance mechanisms.

Similarly, the SE-33 peptide remained undetectable in control groups (0.9% NaCl or native WJ-MSCs), as confirmed by HPLC analysis (Appendix A). In experimental animals, a repeated administration of WJ-MSC-SE33 at a dose of 0.5 × 10^7^ cells/kg resulted in a sustained and tissue-specific distribution of SE-33 (Figure 2, Table 2).

In serum, the peptide reached a C_max_ of 0.233 μg/mL at 8 h post-injection, with an AUC_4–∞_ of 8.05 h×μg/mL and prolonged elimination (T_1/2_ = 23.1 h, λ_z_ = 0.030 h^−1^). The substantial AUMC_4–∞_ (253 h^2^×μg/mL) and extended MRT_4–∞_ (31.4 h) further confirmed extended systemic exposure. Liver tissue exhibited the highest peptide accumulation, with a C_max_ of 1.45 μg/mL (T_max_ 4 h), AUC_4–∞_ of 30.1 h×μg/mL, MRT_4–∞_ of 18.5 h, and T_1/2_ of 12.6 h. In the lungs, the C_max_ was 0.247 μg/mL, with AUC_4–∞_ 3.36 h×μg/mL and a relatively shorter T_1/2_ of 7.67 h, suggesting a more rapid clearance compared to the liver. The spleen exhibited the lowest retention and fastest clearance, with a C_max_ of 0.147 μg/mL and T_1/2_ = 6.80 h.

These findings indicate that the repeated administration of WJ-MSC-SE33 enhances peptide bioavailability and promotes a sustained systemic and organ-specific exposure, particularly in the liver, which may serve as the primary site for peptide metabolism and clearance.

### 2.2. Linear Regression Analysis of the SE-33 Peptide Concentration in Mice Following the Intravenous Administration of WJ-MSC-SE33

To complement the noncompartmental pharmacokinetic analysis, we evaluated the linearity of SE-33 peptide concentration–time profiles across all administered doses of genetically modified WJ-MSCs to mice. A linear regression analysis was conducted at multiple time points (4, 8, 12, 16, 20, 24, 36, and 48 h) following both single and repeated intravenous administration. The strength of the linear fit was quantified by the coefficient of determination (R^2^), and statistical significance was assessed using permutation testing (Table 3, Figure 3A).

The analysis revealed distinct tissue-specific linearity profiles. The liver and serum consistently exhibited the strongest time-dependent linear correlations across all dose groups, while the weakest correlations were observed in the spleen, particularly at the lowest administered dose (0.5 × 10^7^ cells/kg), where SE-33 peptide levels were below the lower limit of quantification (LLOQ). Notably, permutation testing confirmed statistically significant linear relationships (*p* < 0.001) in all tissues, with the exception of the spleen at the lowest dose (R^2^ = 0.069132, *p* = 0.172).

A dose–response trend was evident in the linearity of SE-33 peptide concentrations. At 0.5 × 10^7^ cells/kg, weak-to-moderate correlations were observed in serum (R^2^ = 0.4383) and lungs (R^2^ = 0.6558), while the liver exhibited a strong linear relationship (R^2^ = 0.8704), and the spleen displayed minimal linearity (R^2^ = 0.0691), indicating a very weak correlation. Increasing the dose to 1.25 × 10^7^ cells/kg of WJ-MSC-SE33 improved the linearity in serum (R^2^ = 0.7450) and liver (R^2^ = 0.7857), while lungs remained in the moderate range (R^2^ = 0.5528), and the spleen showed a weak correlation (R^2^ = 0.1991). At the highest administered dose (2.5 × 10^7^ cells/kg), linearity in the spleen modestly increased and improved (R^2^ = 0.4962), suggesting a dose-dependent increase in SE-33 accumulation or retention.

Following the repeated administration of WJ-MSC-SE33 at 0.5 × 10^7^ cells/kg, linear correlation coefficients increased in most tissues, particularly in the spleen (R^2^ = 0.5997), representing a nearly 9-fold enhancement compared to the single administration at the same dose. The liver maintained a strong correlation (R^2^ = 0.8371), while the serum (R^2^ = 0.6580) and lungs (0.5673) exhibited moderate linearity.

Taken together, these findings demonstrate that the SE-33 peptide exhibits a time-dependent pharmacokinetic profile with tissue-specific accumulation dynamics. The strong and dose-independent linearity observed in the liver may suggest a predominant role in peptide metabolism or sequestration. In contrast, serum and lung tissues maintained moderate linearity regardless of dose, while the spleen demonstrated the most dynamic, dose-responsive behavior, potentially reflecting a threshold-dependent saturation of uptake or clearance mechanisms in this secondary lymphoid organ.

### 2.3. The Apparent Distribution and Tissue Penetration of SE-33 Peptide in Mice Following an Intravenous Administration of WJ-MSC-SE33

To further characterize the spatial distribution of the SE-33 peptide, we calculated the tissue penetration factor (fT) and apparent distribution coefficient (Kd) from pharmacokinetic data (Table 4 and Table 5). These parameters were used to evaluate peptide biodistribution following the single and repeated intravenous administration of WJ-MSC-SE33 in mice.

In the liver, a clear dose-dependent decrease in fT values was observed. Compared to the lowest dose (0.5 × 10^7^ cells/kg), fT decreased by 25.4% and 37.5% at 1.25 × 10^7^ and 2.5 × 10^7^ cells/kg, respectively. The repeated administration of 0.5 × 10^7^ cells/kg further reduced fT by 68.2% compared to the single administration at the same dose, indicating substantial saturation or altered tissue retention dynamics with prolonged exposure.

In contrast, lung tissue exhibited an opposite trend under single-dose conditions. The fT values increased modestly with higher doses, by 11.5% and 16.0% at 1.25 × 10^7^ and 2.5 × 10^7^ cells/kg, respectively, compared to the lowest dose. However, the repeated administration of genetically modified WJ-MSC markedly reduced lung fT by 76.3% relative to the single administration at 0.5 × 10^7^ cells/kg, suggesting that repeated dosing may saturate or suppress the pulmonary accumulation of the peptide.

The spleen consistently displayed the lowest fT values among all tissues. At the lowest single dose, fT was not calculable due to concentrations below the LLOQ. At 1.25 × 10^7^ and 2.5 × 10^7^ cells/kg, fT increased from 0.28 to 0.35, indicating a 25.0% rise in peptide penetration. Notably, repeated administration at 0.5 × 10^7^ cells/kg resulted in a fT of 0.42, representing a 20.0% increase relative to the highest single dose, suggesting a cumulative or sensitization effect in splenic tissue.

An analysis of the Kd values over time revealed further organ-specific patterns of distribution. The liver showed consistently high Kd values following single administration, particularly at early time points (4–20 h), reflecting a substantial peptide accumulation. In contrast, Kd values in the lungs were lower but followed similar temporal kinetics. The spleen demonstrated highly variable and low Kd values, which were often not calculable at early time points due to concentrations below the LLOQ, but increased at later time points or following repeated administration, possibly indicating a delayed or limited peptide uptake.

Collectively, these results highlight the complex pharmacokinetic behavior of SE-33 in vivo, demonstrating tissue-specific distribution dynamics influenced by both dose and dosing frequency. The decreasing fT values in the liver and lungs with an increasing dose suggest saturable transport or metabolic processes, while the spleen exhibits a delayed and dose-dependent accumulation, potentially implicating a threshold-mediated uptake or retention mechanisms. These findings on tissue penetration and distribution trends of SE-33 peptide align with the data presented in Figure 3B, which depicts the spatiotemporal distribution of SE-33 across peripheral tissues and serum compartments.

## 3. Discussion

Over the past decade, MSCs have garnered significant attention for their immunomodulatory and antimicrobial properties, positioning them as promising candidates for advanced therapeutic strategies targeting infectious diseases, including sepsis and pneumonia, as well as mitigating organ dysfunction or damage caused by infections [13,14,57,58]. Preclinical and clinical studies have reported that MSC therapy improves patient survival and recovery rates in cases of sepsis or bacterial pneumonia by enhancing the clearance of infectious foci and suppressing bacterial growth [59,60]. Furthermore, MSCs have been shown to increase bacterial sensitivity to antibiotics [27]. In animal models, combination therapy using MSCs and antibiotics has been shown to be highly effective against infections caused by antibiotic-resistant pathogens [61,62]. Despite these promising outcomes, the persistent challenge of antibiotic-resistant pathogens necessitates the development of innovative approaches, including the use of AMPs. AMPs, with their broad spectrum of activity against antibiotic-resistant bacteria and additional roles in immune modulation, angiogenesis, tissue repair, and anticancer effects, represent a promising alternative to conventional antibiotics [63,64,65,66,67,68]. Among AMPs, cathelicidin LL-37 has garnered attention for its effective antibacterial properties and potent role in innate immune responses against bacterial infections [48]. The genetic reprogramming of MSCs to overexpress AMPs, such as LL-37, may hold significant potential for enhancing the therapeutic efficacy of cell-based treatments for bacterial infections.

Recently, we have demonstrated that WJ-MSCs engineered to express the synthetic AMP SE-33 (WJ-MSC-SE33) exhibited a favorable safety profile with no signs of acute or chronic toxicity, organ dysfunction, or pathological alterations in animal models, while also displaying significant antimicrobial effects in a murine model against *S. aureus*-induced pneumonia [56]. These effects included a dose-dependent reduction in bacterial load, the suppression of inflammation, decreased mortality, and an enhanced pathogen clearance compared to native WJ-MSCs, suggesting that SE-33 enhances both the antimicrobial and immunomodulatory functions of MSCs.

In the present study, we characterized the pharmacokinetic profile and tissue distribution of the antimicrobial peptide SE-33 following an intravenous administration of WJ-MSC-SE33 in C57BL/6 mice, aiming to elucidate its tissue-specific distribution, systemic exposure, and elimination profile after single and repeated dosing. Peak SE-33 concentrations were observed in the lungs within 4 h post-administration (0.233–0.352 μg/mL), while peak levels in the liver (0.965–1.428 μg/mL), serum (0.142–0.262 μg/mL), and spleen (0.044–0.133 μg/mL) were detected at 8 h consistent with the initial pulmonary retention of WJ-MSCs followed by systemic redistribution. Notably, SE-33 remained detectable in the liver for up to 48 h at all dose levels, with high AUC values (24.0 to 38.0 h×μg/mL) and a prolonged MRT (20.4 to 21.9 h), suggesting a hepatic accumulation and possible metabolism.

The lungs demonstrated a dose-dependent retention of SE-33, with detectability extending up to 48 h at medium and high doses and up to 20 h at the lowest dose. The elimination half-life increased from 4.34 to 22.9 h across the dosing range, highlighting a dose-dependent saturation of pulmonary clearance mechanisms. Serum pharmacokinetics exhibited intermediate characteristics, with dose-proportional increases in AUC (2.04–5.17 h×μg/mL) and stable half-life values (4.00–4.80 h). In contrast, the spleen showed a limited penetration, with detectable levels only at medium and high doses (AUC 1.03–1.81 h×μg/mL) and the rapid elimination of SE-33, likely due to either poor accumulation or rapid degradation.

In addition to classical non-compartmental analysis, linear regression modeling was employed to further characterize the temporal pharmacokinetic behavior of SE-33 across tissues. Notably, the liver exhibited a strong, dose-independent linear relationship (R^2^ > 0.77 across all dosing regimens), suggesting a potential predominant role in SE-33 peptide metabolism or sequestration. In contrast, serum and lung tissues demonstrated moderate linearity regardless of dose, whereas the spleen displayed a dose-dependent response characterized by dynamic accumulation patterns, reflecting a threshold-dependent saturation of peptide uptake or clearance mechanisms.

Although linear regression remains a widely used approach for estimating elimination parameters during the terminal phase of pharmacokinetics, it carries several limitations [69,70]. In particular, this method assumes a log-linear decay of concentration over time and is sensitive to deviations from this assumption, that may arise due to multi-compartmental kinetics, nonlinear tissue distribution, or heterogeneity in cellular uptake rates. These challenges are especially relevant in the context of cell-based delivery platforms such as MSCs, where the dynamics of molecule release, cellular trafficking, and tissue-specific homing introduce complexities that are not fully captured by linear models [71,72]. Moreover, the application of linear regression in this study does not account for inter-individual variability or biological heterogeneity, which may be more appropriately addressed through nonlinear mixed-effects modeling in future investigations. Despite these limitations, the regression-based analysis in our study supported the tissue-specific pharmacokinetic trends identified via non-compartmental analysis and contributed to the identification of organs exhibiting the preferential accumulation of the SE-33 peptide.

These pharmacokinetic characteristics reveal an organ-specific distribution pattern of SE-33. The liver emerged as the major reservoir and potential site of elimination, while lung retention was modulated primarily by administered cell dose. These observations on the accumulation of SE-33 in these organs likely associated with the properties of both the intravenously administered WJ-MSCs and cathelicidins themselves [73,74,75,76].

Previous studies have demonstrated that both bone marrow-derived and umbilical cord-derived MSCs predominantly accumulate in the lungs due to the pulmonary first-pass effect, with smaller amounts detected in the liver and kidneys within the first 24 h following intravenous administration [73,75,76,77,78,79,80]. Similarly, WJ-MSCs exhibit a rapid pulmonary accumulation, followed by clearance within 24–72 h [81,82,83]. Specifically, WJ-MSCs initially localize in the lungs and liver, with a significant retention in the pulmonary tissue at 2–4 h, after which they redistribute to other organs, such as kidneys, spleen, and heart, within 24–48 h [81,82]. By 48 h, xenogeneic WJ-MSCs in the lungs and blood significantly decrease or become undetectable, while their presence in the liver increases over time [83]. This pattern of biodistribution likely explains the shorter half-life of SE-33 peptide in the pulmonary tissues observed in our study. Notably, WJ-MSCs can still be detected in the spleen up to 72 h following intravenous administration, although in significantly lower quantities than in other tissues [83]. Moreover, human umbilical cord-derived MSCs may persist for at least 14 days in multiple tissues, including the kidneys [78]. Consistent with these observations, we detected the lowest concentration of SE-33 in the spleen. Given that WJ-MSCs can still be detected in this organ at later time points, this suggests either a reduced initial accumulation of WJ-MSC-SE33 in the spleen or a more rapid degradation of the SE-33 peptide in this tissue [83]. Taken together, these findings support our data on the biodistribution of SE-33 in various tissues following the intravenous administration of WJ-MSC-SE33.

Interestingly, although the SE-33 concentrations measured in vivo in the present study were below the minimum inhibitory concentration (MIC) (31.2–1024 μg/mL for fungi and yeast; 12.5 μg/mL for *S. aureus*) and the minimum bactericidal concentration (MBC) (20 μg/mL for *E. coli*) thresholds established in vitro, substantial antimicrobial effects were previously observed in animal models [53,55,56]. These findings suggest that the in vivo efficacy of SE-33 is not solely dependent on achieving bactericidal concentrations, but may instead rely on prolonged tissue retention, local immunomodulation, and the cumulative exposure afforded by a sustained peptide release. The extended half-lives and high AUC values support this notion, as do the immunosuppressive and inflammatory modulating effects observed in treated animals [56].

Moreover, we previously quantified the peptide content of WJ-MSC-SE33 cells, estimating 0.8 μg of SE-33 peptide per 1 × 10^5^ cells [56]. Given the administration of 1 × 10^5^ to 5 × 10^5^ cells per injection, the total peptide doses ranged from 0.8 to 4.0 μg, corresponding to sub-MIC levels. Nonetheless, this dosing regimen achieved a 12-fold reduction in the bacterial load in bronchoalveolar lavage fluid by day 5 post-injection, sustaining a 2-fold reduction through day 8 [56]. These findings imply that the in vivo action mechanism of SE-33 involves more than direct bactericidal activity and may include immune-mediated antibacterial responses, possibly through the modulation of macrophage or neutrophil activity.

Interestingly, the highest concentration of the SE-33 peptide was detected in the liver tissue rather than in the lungs, with levels exceeding those in the lungs, 4-fold. This observation may be attributed to the role of the liver in the clearance and metabolism of the SE-33 peptide. A recent study demonstrated that cathelicidin LL-37 is rapidly eliminated from the bloodstream primarily by the liver, even though a significant fraction temporarily accumulates in the lungs during the first 48 h following intravenous administration [74]. Our findings on the biodistribution of SE-33 peptide in body tissues after intravenous administration are consistent with these observations. Furthermore, the lower levels of SE-33 in the lungs compared to the liver may be attributed to the activity of lung peptidases, which are produced by bronchial epithelial cells and alveolar macrophages [84,85]. For example, cathepsins K and S found in the lungs can cleave LL-37 in the mid-region at cleavage sites such as Gln22-Arg23, Lys25-Asp26, and Arg29-Asn30 [86]. This enzymatic activity may lead to the formation of truncated forms of AMPs, including LL-37 and SE-33, or diminish their biological activity through proteolysis in the active core region (residues 17–29) [36,53,87]. Therefore, the chromatography method we developed to quantify SE-33 peptide levels may have limitations, as degraded or truncated peptide fragments might remain undetected.

The repeated administration of WJ-MSC-SE33 at a minimal dose produced a pharmacokinetic profile distinct from the single administration. Four hours after the final injection, SE-33 was detectable in both lungs and liver, with elevated peptide levels in the spleen compared to single-dose administration. This increase suggests either an enhanced systemic accumulation or cumulative peptide release over time.

Notably, the liver showed a dose-dependent decrease in both the apparent distribution coefficient and tissue penetration factor with increasing WJ-MSC-SE33 doses, indicating a potential saturation of hepatic uptake or degradation pathways. Among the peripheral organs, the liver and lungs were identified as the primary sites of SE-33 accumulation, with the liver showing the highest tissue penetration factor and apparent distribution coefficient values. In contrast, the distribution coefficient for SE-33 in the spleen following a single intravenous administration was significantly lower compared to other organs, tending toward zero. This observation may indicate either a limited accumulation or rapid elimination of the SE-33 from spleen tissues. The high tissue penetration values observed in the liver suggest that hepatic excretion may be a significant route of SE-33 elimination for intravenously administered WJ-MSC-SE33 [74,88,89]. However, the decrease in liver tissue penetration with increasing doses of WJ-MSC-SE33 suggests a saturation effect or limited capacity of the liver for further SE-33 peptide accumulation.

With repeated administrations of WJ-MSC-SE33, both the apparent distribution coefficient and the tissue penetration factor of SE-33 were significantly decreased in the liver and lungs. Specifically, the apparent distribution coefficient of SE-33 decreased approximately 3-fold in the liver and 4-fold in the lungs. This reduction likely indicates the elimination or metabolism of accumulated SE-33 in the liver between administrations. Furthermore, this phenomenon may reflect adaptive tissue responses or limitations in hepatocellular transport and metabolism.

Taken together, our data highlight the influence of both the dose and frequency of WJ-MSC-SE33 administration on SE-33 pharmacokinetics. The liver and lungs consistently emerged as the primary organs for peptide retention. While hepatic accumulation suggests a major route of clearance, the sustained presence in lungs, our primary target organ, supports the therapeutic feasibility of MSC-based AMP delivery for pulmonary infections. The decreased tissue penetration observed upon repeated dosing may be due to the peptide clearance between injections or the saturation of the MSC homing and secretion capacity. The findings underscore the importance of optimizing dosing regimens to balance therapeutic efficacy with tissue-specific pharmacokinetics.

## 4. Materials and Methods

### 4.1. Plasmid Vector Construction and Mesenchymal Stromal/Stem Cell Transfection

The plasmid vector pVIDBse33 (Figure 4A) was constructed, produced, and purified following previously described protocols [53,54,56]. Upon the transfection of MSCs, this plasmid expresses the SE-33 peptide, with the amino acid sequence SETRPVLNRLFDKIRQVRKEFGKIKEKSRKFM.

Human Wharton’s Jelly-derived mesenchymal Stromal/Stem cells (WJ-MSCs) were obtained from the Cell Culture Collections for Biotechnological and Biomedical Research at the Koltzov Institute of Developmental Biology of The Russian Academy of Sciences (Moscow, Russia). The isolation, characterization, identification, trilineage differentiation, and storage of WJ-MSCs were then documented in the cell passport, available at https://www.en.idbras.ru/index.php/en/institute/bio-resources/88-cell-culture-collection (accessed on 15 December 2024). The cells used in this study are designated as WJMSC (20200528).

WJ-MSCs were cultured and transfected with the constructed plasmid as previously described [56]. The resulting genetically modified WJ-MSCs expressing the SE-33 peptide were designated as WJ-MSC-SE33. WJ-MSC-SE33 were resuspended in 0.9% NaCl for subsequent animal studies. Both native WJ-MSCs and WJ-MSC-SE33, along with a culture medium (supernatants), were subjected to an antimicrobial activity assessment prior to use as previously described [56].

### 4.2. Animal Studies

The studies were conducted on sexually mature male C57BL/6 mice with an average body weight of 20 g and an age of 8 weeks. The animals were sourced from the “Andreevka”, branch of the Scientific Center of Biomedical Technologies of the Russian Federal Medical and Biological Agency (Russia). The mice underwent a 14-day acclimatization period. The animals were housed in individually ventilated polysulfone cages ISO RAIR 4 × 8 (LabProducts, Aberdeen, WA, USA) with Rehofix MK2000 corn cob bedding (J. Rettenmaier and Söhne GmbH, Rosenberg, Germany). The animal housing environment followed a 12-h light/dark cycle with an ad libitum access to water and standard laboratory feed. The group assignment was based on body weight, ensuring a no more than 10% deviation from the average group weight. All experimental procedures were approved by the Local Ethics Committee (LEC) of the Federal State Budgetary Educational Institution of Higher Education “Privolzhsky Research Medical University” Ministry of Health of the Russian Federation (Protocol No. 06, dated 29 April 2022) and adhered to the guidelines for the care and use of laboratory animals [90].

### 4.3. The Administration of Genetically Modified WJ-MSCs

Immediately prior to administration, WJ-MSC-SE33 cells were resuspended in a saline solution (0.9% NaCl). The mice were divided into groups (n = 48) and fasted for 8 h before intravenous injection. The injection volume was calculated based on body weight. Single-dose administrations were 1 × 10^5^ cells/mouse (0.5 × 10^7^ cells/kg), 2.5 × 10^5^ cells/mouse (1.25 × 10^7^ cells/kg), and 5 × 10^5^ cells/mouse (2.5 × 10^7^ cells/kg) (Figure 4B). Repeated administrations were conducted on days 1, 3, and 5 at a dose of 1 × 10^5^ cells/mouse (Figure 4B).

The maximum administered dose of WJ-MSC-SE33 cells was determined based on safety considerations to avoid vascular complications. The maximum tolerated dose of 5 × 10^5^ cells/mouse (2.5 × 10^7^ cells/kg) in a volume of 0.2 mL 0.9% NaCl was determined in preliminary studies, as higher doses or injection volumes induced vascular complications, including clot formation and hemodynamic disturbances, unrelated to drug toxicity [91]. The dosing regimen for single and repeated administrations was selected in accordance with the Decision of the College of the Eurasian Economic Commission No. 202 (dated 26 November 2019; amended 11 October 2022) “On Approval of the Guidelines for Preclinical Safety Studies for the Purpose of Conducting Clinical Trials and Registering Medicinal Products”, which align with the International Council for Harmonization of Technical Requirements for Pharmaceuticals for Human Use [92,93].

### 4.4. Biomaterial Collection

Blood samples were collected from the subclavian artery at 4, 8, 12, 16, 20, 24, 36, and 48 h post-administration (n = 6 animals/time point) (Figure 1B). Based on preliminary data showing no detectable SE-33 levels within 2 h post-intravenous administration of WJ-MSC-SE33 (2.5 × 10^7^ cells/kg, n = 9), and this time interval was excluded from sample collection (Appendix A). The 0 h time point represents samples obtained from animals prior to WJ-MSC-SE33 injection (pre-dose baseline). SE-33 concentrations at baseline were established as 0 μg/mL since the peptide was undetectable (<LLOQ) in all pre-dose samples, confirming the absence of endogenous expression. Blood samples were collected into tubes without an anticoagulant and centrifuged at 3000 rpm for 15 min at room temperature. Then, serum samples were isolated and placed in cryovials. Samples of peripheral tissues (lungs, liver, and spleen) were collected at similar time points. Blood and tissue samples were stored at −79 °C.

### 4.5. The Determination of the Antimicrobial Peptide SE-33 in Animal Tissue and Serum Samples

#### 4.5.1. Sample Preparation for Chromatographic Analysis

Thawed mouse serum was centrifuged at 12,000 rpm for 30 min using a MiniSpin centrifuge (Eppendorf, Hamburg, Germany). The supernatant was collected and diluted 20-fold with 20 mM TrisHCl (pH 7.8). For peripheral tissue samples, 100 mg of thawed tissue was weighed using Sartorius CPA225D scales (Sartorius, Göttingen, Germany) and homogenized in 1.6 mL of acidified ethanol (0.5 mL HCl in 1000 mL ethanol) using a Heidolph Diax 100 homogenizer (Heidolph Instruments, Schwabach, Germany) at 25,000 rpm on ice for 1 min. The homogenate was centrifuged at 12,000 rpm for 30 min. The supernatant was collected and diluted 20-fold with 20 mM Tris-HCl (pH 7.8).

#### 4.5.2. The Preparation of an Affinity Column for SE-33 Peptide Binding

To construct a protein that specifically binds to the SE-33 peptide (245 amino acids in length), a complementary sequence with a high affinity to the SE-33 was calculated using the Antimicrobial Peptide Database APD3 [94,95]. The sequence was cloned into a plasmid and expressed in *E. coli* strain BL21(DE3). The protein was then isolated and purified using a Ni-NTA column. Purified protein was conjugated with Epoxy-activated Sepharose 6B, creating a sorbent (Epoxy-Protein Sepharose) with a binding capacity of 240 μg/mL for the SE-33 peptide.

#### 4.5.3. Affinity Chromatography

A column containing 5 mL of Epoxy-Protein Sepharose was equilibrated with 20 mM TrisHCl (pH 7.8). The sample was applied to the column via a 2 mL loop, followed by washing with 25 mL of 20 mM TrisHCl (pH 7.8). The SE-33 peptide was eluted with 5 mL of 75 mM NaCl in 20 mM TrisHCl (pH 7.8) without fractionation. The eluate enriched with the SE-33 peptide was subjected to solid-phase extraction.

#### 4.5.4. Solid-Phase Extraction

The eluate (5 mL) obtained from affinity chromatography was diluted with 20 mL of 0.1% trichloroacetic acid (TCA) and applied to a Chromabond C18 Hydra 6 mL/500 mg column cartridge. The column was washed with 30 mL of 0.1% TCA in 5% acetonitrile, and the SE-33 peptide was eluted with 90% acetonitrile. The eluate was prepared for high-performance liquid chromatography (HPLC) analysis.

#### 4.5.5. High-Performance Liquid Chromatography

The eluate was vacuum-evaporated using a Savant SC-210A vacuum centrifuge (Savant, Thermo Scientific, Waltham, MA, USA) and dissolved in 0.1% TCA in 5% acetonitrile. HPLC was performed using a Kromasil Ambercrome 100-5-C18 column (4.6 × 250 mm) on a Breeze GA chromatograph (Waters, Milford, KT, USA). The column was equilibrated with Eluent A (0.1% TCA in 5% acetonitrile) and Eluent B (0.1% TCA in 65% acetonitrile). The mobile phase flow rate was set at 2.0 mL/min, and detection was performed at 228 nm. A total of 20 μL of the sample was used for analysis.

The concentration of the free fraction of the SE-33 peptide in the experimental samples was quantified by HPLC analysis using peak area integration relative to a calibration curve generated with SE-33 standard solutions, as previously described [56]. Representative HPLC profiles and the corresponding calibration curve used for method validation are presented in Appendix A. The SE-33 peptide for standard solution preparation was synthesized and purified according to previously established protocols [53,54]. The LLOQ for SE-33 detection in samples was determined to be 0.10 μg/mL.

### 4.6. Non-Compartmental Analysis of Pharmacokinetic Data

Pharmacokinetic parameters were calculated through noncompartmental analysis (NCA). NCA was performed using the R software (version 4.4.2, R Foundation for Statistical Computing, Vienna, Austria) with the PKCNA package [96,97], which provides an automated and reliable framework for calculating pharmacokinetic parameters, including the following:

C_max_ (μg/mL): the maximum observed concentration of SE-33 in serum or tissues.

T_max_ (h): the time to reach C_max_. This was measured at 4, 8, 12, 16, 20, 24, 36, and 48 h post-administration. T = 0 was set as the baseline (0 μg/mL, measurement performed before administration).

λ_z_ (h^−1^): terminal elimination rate constant.

t_1/2_ (h): elimination half-life, calculated as follows:(1)t12=ln(2)λz

AUC_0–t_ (h×μg/mL): the area under the concentration–time curve from administration to the last measurable timepoint (24 or 48 h).

AUC_0–∞_ (h×μg/mL): AUC extrapolated to infinity:(2)$${AUC}_{0–\infty }={AUC}_{0–t}+\frac{C_{last}}{\lambda_{z}}$$
where C_last_ is the last quantifiable concentration.

AUMC_0–t_ (μg×h^2^/mL): the area under the first moment curve (concentration × time versus time).

MRT (h): mean residence time:(3)$$MRT=\frac{{AUMC}_{0–t}}{{AUC}_{0–t}}$$

### 4.7. Tissue Penetration and Distribution Analysis

The tissue availability and distribution of SE-33 peptide were further assessed using the following parameters:

fT: tissue penetration factor. This parameter reflects the ratio of drug exposure in tissues relative to plasma or serum and is used to assess antimicrobial drug distribution at the target site and to guide therapy selection [98]. In this study, fT was calculated using the following equation:(4)$$fT=\frac{{AUC}_{0–t(tissue)}}{{AUC}_{0–t(serum)}}$$

This approach, commonly applied in evaluating drug distribution within extravascular compartments, particularly for antimicrobial agents, is consistent with its established definition in pharmacokinetics as the tissue-to-plasma AUC ratio [99,100]. Values of fT < 0.2 may be interpreted as indicative of limited tissue penetration [101].

Kd: apparent distribution coefficient. This parameter was calculated during the mono-exponential phase as the ratio of tissue to serum concentration. Values of Kd > 1 indicate tissue accumulation.(5)$$Kd=\frac{C_{tissue}}{C_{serum}}$$

### 4.8. Linear Regression Analysis

Linear regression analysis was performed to evaluate the goodness-of-fit for concentration–time relationships using built-in functions in R software (version 4.4.2, R Foundation for Statistical Computing, Vienna, Austria). This approach provided a quantitative measure of the linear association between time and SE-33 peptide concentration across tissues.

Data processing was implemented with the dplyr (v1.1.4) and the tidyverse (v1.3.0) packages to facilitate data transformation, summarization, and aggregation [96,102,103]. Visualization was performed using ggplot2 package (v3.3.3) [104].

### 4.9. Statistical Analysis

Data were analyzed using IBM SPSS Statistics 26 software (IBM, Armonk, New York, NY, USA) and R software (version 4.4.2, R Foundation for Statistical Computing, Vienna, Austria) [96]. Descriptive statistics included the mean (M) and standard deviation (SD) or standard error of the mean (SEM). The normality of data distribution in the groups was assessed using the Shapiro–Wilk test. One-way ANOVA with Bonferroni’s post hoc correction was used for multiple comparisons. Differences between groups were considered statistically significant at a *p*-value of *p* < 0.05.

## 5. Conclusions

In this study, we investigated the pharmacokinetics of the antimicrobial peptide SE-33 expressed by genetically modified WJ-MSCs (WJ-MSC-SE33) following single and repeated intravenous administration to C57BL/6 mice. Our findings demonstrate that the biodistribution and persistence of SE-33 in systemic circulation and major organs are dose-dependent, with the highest peptide concentrations observed in the lungs and liver. Peak levels were detected in the lungs at 4 h post-injection, while maximal serum and hepatic concentrations occurred at 8 h. The peptide remained detectable in the blood for up to 36 h and in tissues for up to 48 h following administration, particularly at higher cell doses.

Pharmacokinetic parameters revealed that the liver and lungs serve as key compartments for SE-33 retention and elimination, with the liver demonstrating the longest peptide half-life and highest AUC values. The relatively lower SE-33 concentrations in the spleen may reflect a reduced penetration or rapid peptide degradation. The accumulation pattern of SE-33 in tissues and organs is likely associated with the distribution and metabolism of both the antimicrobial peptide and the WJ-MSCs. Notably, increasing the administered dose of WJ-MSC-SE33 resulted in higher SE-33 concentrations and a prolonged systemic retention, indicating a dose-dependent distribution profile of the peptide. The repeated administration of WJ-MSC-SE33 resulted in a sustained SE-33 accumulation in the liver and lungs, supporting the potential of cumulative exposure as a mechanism underlying the in vivo efficacy of the peptide. Our findings indicate that WJ-MSC-SE33 provided a targeted and temporally sustained release of SE-33 in the lungs—the primary site of respiratory infections—and in the liver, which likely serves as the main organ for peptide clearance.

These findings demonstrate that genetically modified WJ-MSCs expressing the SE-33 peptide possess predictable pharmacokinetic properties and may serve as a promising basis for the development of novel treatment strategies for infectious diseases. However, further studies are required to assess the long-term safety of this therapy, particularly in the context of chronic infections. It is particularly important to emphasize the potential of genetically modified MSCs expressing AMPs in combating infections complicated by the antibiotic resistance of bacteria. Future studies should focus on assessing the effectiveness of combining this cell-based therapy with antibiotic treatments to enhance bactericidal activity against both Gram-positive and Gram-negative bacteria. This approach is particularly critical in the context of the global rise of antibiotic resistance. Moreover, to deepen our understanding of the biodistribution and fate of WJ-MSC-SE33 following administration, future studies should employ advanced cell-tracking methodologies to monitor the migration, retention, and clearance of transplanted cells in vivo. Such approaches would allow for a more precise correlation between cell localization and peptide release dynamics, helping to clarify the contribution of MSC viability, homing behavior, and local microenvironmental interactions to therapeutic outcomes.

## Figures and Tables

**Figure 1 antibiotics-14-00429-f001:**
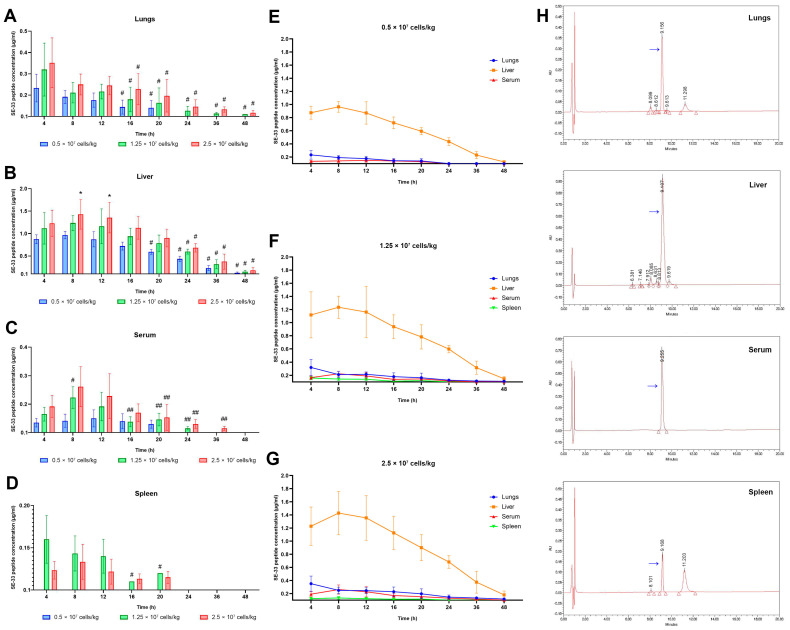
The detection of SE-33 peptide in serum and peripheral tissues of mice after a single administration of WJ-MSC-SE33 at different doses. The SE-33 peptide concentration in the lungs (**A**), liver (**B**), serum (**C**), and spleen (**D**) of mice after a single administration of WJ-MSC-SE33 at different doses, measured at each time point. One-way ANOVA with Bonferroni’s correction, M ± SD. # *p* < 0.05 compared with the SE-33 peptide concentration in samples collected at the 4-h time point. ## *p* < 0.05 compared with the SE-33 peptide concentration in samples collected at the 8-h time point. * *p* < 0.05 compared with SE-33 peptide concentration in tissue samples following the administration of 0.5 × 10^7^ cells/kg at each time point. Panels (**E**–**G**) demonstrate time-dependent changes in the concentration of SE-33 peptide in the lungs, liver, serum, and spleen following a single administration of WJ-MSC-SE33 at doses of 0.5 × 10^7^, 1.25 × 10^7^, and 2.5 × 10^7^ cells/kg. Panel (**H**) shows representative HPLC profiles of the lungs, liver, spleen, and serum samples from mice following a single intravenous administration of WJ-MSC-SE33. Blue arrows indicate peak concentrations corresponding to the SE-33 peptide. In the lungs, the SE-33 peptide was detected at a concentration of 0.26 μg/mL, with a retention time of 9.156 min. In the liver, the SE-33 peptide concentration was 1.40 μg/mL, with a retention time of 9.107 min. In blood serum, the SE-33 peptide concentration was 0.35 μg/mL, with a retention time of 9.225 min. In the spleen, the SE-33 peptide concentration was 0.16 μg/mL, with a retention time of 9.168 min.

**Figure 2 antibiotics-14-00429-f002:**
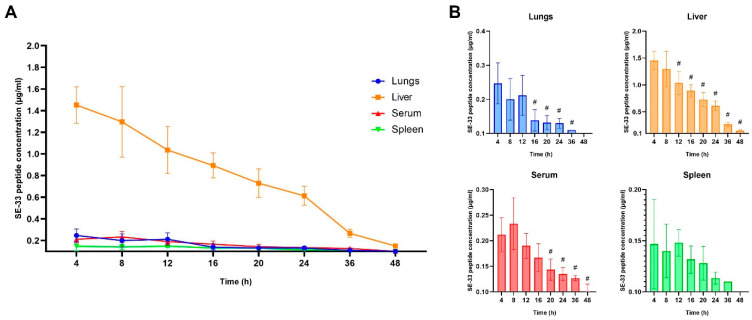
Pharmacokinetics of SE-33 peptide expressed by WJ-MSC-SE33 in mice following repeated intravenous administration at a dose of 0.5 × 10^7^ cells/kg. (**A**) Time-dependent changes in the concentration of SE-33 peptide in the blood serum and organ tissues of mice after repeated administration. (**B**) SE-33 peptide concentrations in peripheral tissues and blood serum at each time point. One-way ANOVA with Bonferroni’s post hoc correction, M ± SD. # *p* < 0.05 compared with the SE-33 peptide concentration at the 4-h time point.

**Figure 3 antibiotics-14-00429-f003:**
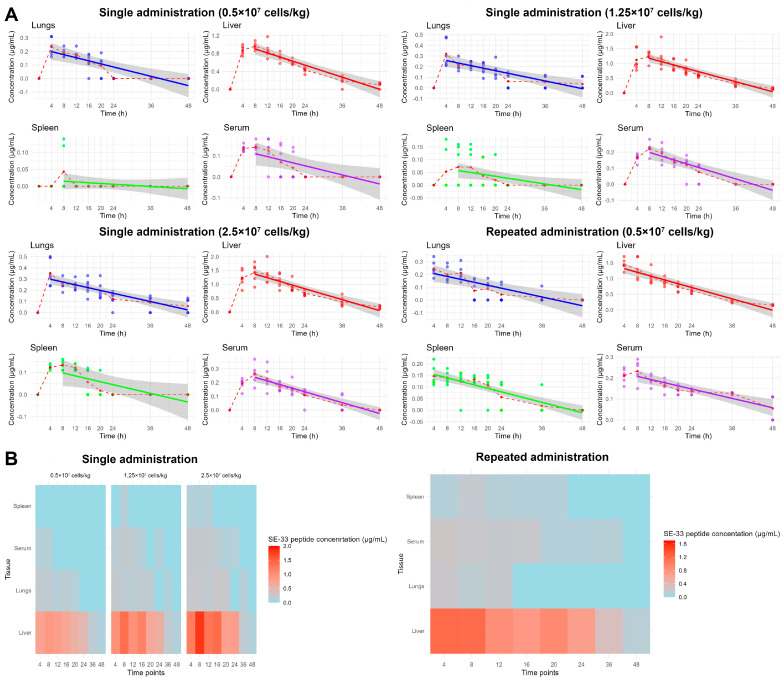
The linear regression analysis and tissue distribution of the SE-33 peptide following the administration of WJ-MSC-SE33 in mice. (**A**) The linear time dependence of the SE-33 peptide concentration in serum and organ tissues of mice following the single and repeated intravenous administration of the WJ-MSC-SE33. The dots on the graph represent individual measurements of peptide concentration at each time point. The red dashed lines connect the mean values of peptide concentration at each time point, represented by red dots. These dashed lines illustrate the overall distribution trend of the observed data across time. The colored solid lines represent the results of linear regression models fitted to the mean values (blue for lungs, red for liver, green for spleen, and purple for serum). The gray shaded areas surrounding the regression lines denote the 95% confidence intervals, reflecting the uncertainty associated with the fitted linear models. (**B**) The spatial distribution of the SE-33 peptide in peripheral tissues and blood serum at each time point.

**Figure 4 antibiotics-14-00429-f004:**
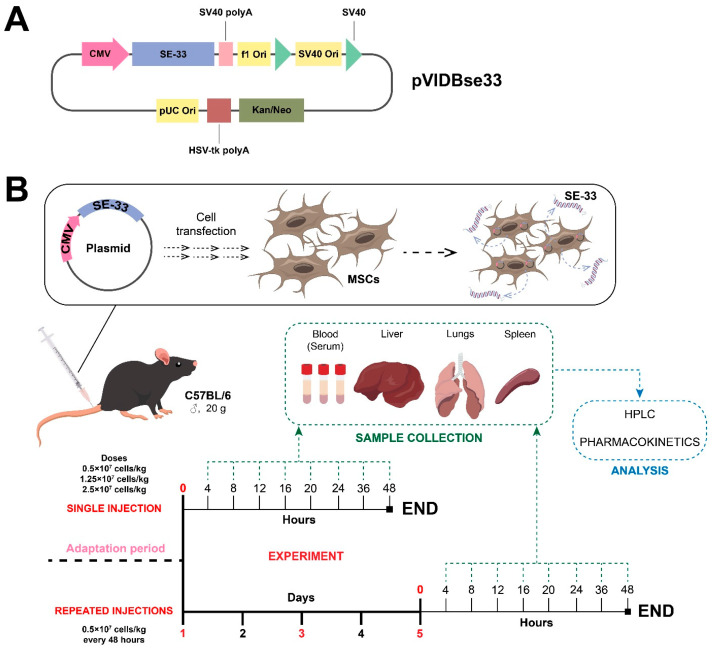
The experimental design for the pharmacokinetic study of genetically modified WJ-MSC. (**A**) A map of the developed plasmid vector. (**B**) Administration conditions for genetically modified WJ-MSC expressing the antimicrobial peptide SE-33 in an animal model.

**Table 1 antibiotics-14-00429-t001:** Pharmacokinetic parameters in the peripheral tissues and serum of mice after a single administration of genetically modified WJ-MSC.

Parameter	Serum			Lungs			Liver			Spleen		
	1	2	3	1	2	3	1	2	3	1	2	3
C_max_ (μg/mL)	0.142	0.223	0.262	0.233	0.320	0.352	0.965	1.235	1.428	0.044	0.072	0.133
T_max_ (h)	8	8	8	4	4	4	8	8	8	8	8	8
λ_z_ (h^−1^)	0.144	0.174	0.173	0.160	0.048	0.031	0.061	0.057	0.056	NC	0.124	0.136
T_1/2_ (h)	4.80	3.98	4.00	4.34	14.3	22.9	11.4	12.2	12.4	NC	5.59	5.09
AUC_0–24_ (h×μg/mL)	2.01	4.58	6.08	3.15	5.13	6.86	24.6	33.1	36.4	NC	0.999	1.78
AUC_4–24_ (h×μg/mL)	1.74	4.25	5.70	2.68	4.48	6.15	22.8	30.9	34.0	NC	0.889	1.53
AUC_0–∞_ (h×μg/mL)	2.04	3.73	5.17	3.18	6.50	9.37	24.0	32.7	38.0	NC	1.03	1.81
AUC_4–∞_ (h×μg/mL)	1.77	3.4	4.79	2.71	5.86	8.67	22.2	30.4	35.6	NC	0.918	1.56
AUMC_0–24_ (h^2^×μg/mL)	20.0	82.3	127	31.7	73.3	119	519	733	746	NC	10.4	16.9
AUMC_4–24_ (h^2^×μg/mL)	12.0	64.0	103	19.2	52.8	91.4	421	601	600	NC	6.41	9.80
AUMC_0–∞_ (h^2^×μg/mL)	21.2	47.8	79.7	32.9	148	292	489	702	833	NC	11.6	18.1
AUMC_4–∞_ (h^2^×μg/mL)	13.1	32.9	59.1	20.2	122	255	393	571	680	NC	7.53	10.9
MRT_0–24_ (h)	9.95	18.0	20.9	10.1	14.3	17.3	21.1	22.1	20.5	NC	10.4	9.51
MRT_4–24_ (h)	6.89	15.1	18.1	7.15	11.8	14.9	18.4	19.4	17.7	NC	7.21	6.41
MRT_0–∞_ (h)	10.4	12.8	15.4	10.4	22.7	31.2	20.4	21.5	21.9	NC	11.3	10.0
MRT_4–∞_ (h)	7.38	9.68	12.3	7.46	20.8	29.4	17.7	18.8	19.1	NC	8.20	6.99

Notes: 1—dose of 0.5 × 10^7^ cells/kg; 2—dose of 1.25 × 10^7^ cells/kg; 3—dose of 2.5 × 10^7^ cells/kg; NC—not calculable.

**Table 2 antibiotics-14-00429-t002:** Pharmacokinetic parameters in serum and peripheral tissues after the repeated administration of WJ-MSC-SE33.

Parameter	Serum	Lungs	Liver	Spleen
C_max_ (μg/mL)	0.233	0.247	1.45	0.147
T_max_ (h)	8	4	4	4
λ_z_ (h^−1^)	0.030	0.091	0.055	0.102
T_1/2_ (h)	23.1	7.67	12.6	6.80
AUC_4–24_ (h×μg/mL)	8.21	3.25	33.0	2.94
AUC_4–∞_ (h×μg/mL)	8.05	3.36	30.1	2.99
AUMC_4–24_ (h^2^×μg/mL)	281	34.1	708	37.5
AUMC_4–∞_ (h^2^×μg/mL)	253	38.3	556	39.3
MRT_4–24_ (h)	34.2	10.5	21.5	12.8
MRT_4–∞_ (h)	31.4	11.4	18.5	13.2

**Table 3 antibiotics-14-00429-t003:** Linearity of the SE-33 peptide concentration changes following the single and repeated administration of genetically modified WJ-MSC in mice.

Administration, Dose	Coefficient of Determination (R^2^)			
	Serum	Lungs	Liver	Spleen
Single, 0.5 × 10^7^ cells/kg	0.438344	0.655836	0.870367	0.069132
Single, 1.25 × 10^7^ cells/kg	0.744957	0.552811	0.785668	0.199126
Single, 2.5 × 10^7^ cells/kg	0.723056	0.581298	0.776045	0.496153
Repeated, 0.5 × 10^7^ cells/kg	0.658007	0.567349	0.837087	0.599679

**Table 4 antibiotics-14-00429-t004:** The tissue penetration factor (fT).

Administration, Dose	Liver	Lungs	Spleen
Single, 0.5 × 10^7^ cells/kg	11.76	1.56	NC
Single, 1.25 × 10^7^ cells/kg	8.77	1.74	0.28
Single, 2.5 × 10^7^ cells/kg	7.35	1.81	0.35
Repeated, 0.5 × 10^7^ cells/kg	3.74	0.37	0.42

Notes: NC—not calculable.

**Table 5 antibiotics-14-00429-t005:** The apparent distribution coefficient (Kd).

Administration, Dose	Time (h)	Liver	Lungs	Spleen
Single, 0.5 × 10^7^ cells/kg	4	6.48	1.73	NC
	8	6.80	1.35	0.30
	12	6.98	1.42	NC
	16	10.31	1.71	NC
	20	13.79	2.16	NC
	24	NC	NC	NC
	36	NC	NC	NC
	48	NC	NC	NC
Single, 1.25 × 10^7^ cells/kg	4	6.78	1.94	0.32
	8	5.54	0.95	0.32
	12	6.05	1.13	0.36
	16	6.80	1.30	0.27
	20	6.42	1.34	0.16
	24	7.77	0.82	NC
	36	NC	NC	NC
	48	NC	NC	NC
Single, 2.5 × 10^7^ cells/kg	4	6.40	1.83	0.64
	8	5.45	0.95	0.51
	12	5.94	1.07	0.54
	16	5.93	1.20	0.30
	20	5.90	1.29	0.12
	24	6.32	1.13	NC
	36	9.87	2.32	NC
	48	NC	NC	NC
Repeated, 0.5 × 10^7^ cells/kg	4	1.06	1.17	0.69
	8	1.11	0.86	0.60
	12	1.11	1.12	0.65
	16	1.09	0.44	0.79
	20	1.20	0.62	0.75
	24	1.07	0.32	0.42
	36	0.97	0.13	0.15
	48	1.36	NC	NC

Notes: NC—not calculable.

## Data Availability

The original contributions presented in this study are included in the article and Appendix A. Further inquiries can be directed to the corresponding author.

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
