# Peer review of "Genetically Modified Mesenchymal Stromal/Stem Cells as a Delivery Platform for SE-33, a Cathelicidin LL-37 Analogue: Preclinical Pharmacokinetics and Tissue Distribution in C57BL/6 Mice"

_antibiotics, 2025, doi:10.3390/antibiotics14050429_

Round 1
Reviewer 1 Report
Comments and Suggestions for Authors
The authors describe the pharmacokinetic properties and tissue distribution of a novel therapeutic—Wharton’s jelly–derived mesenchymal stromal/stem cells (MSCs) expressing the antimicrobial peptide SE-33 (WJ-MSC-SE33). Mesenchymal stromal/stem cells have recently garnered interest as both therapeutic agents and delivery platforms. In this study, the authors utilized Wharton’s Jelly–derived MSCs to deliver the antibacterial peptide SE-33, presenting plasma concentration and tissue distribution data following intravenous administration. While this approach is promising, there are several major concerns that should be addressed:
- Please clarify why sample collection was performed at 4 hours post–IV administration and how pharmacokinetic parameters were calculated when including the 0-hour time point.
- Because the MSCs themselves serve as the delivery platform for SE-33, it would be beneficial to present the pharmacokinetic properties of the MSCs to elucidate the release profile of SE-33. In the Discussion (page 12, line 374), the authors attribute the lowest concentration of SE-33 in the spleen to either reduced accumulation of WJ-MSC-SE33 or rapid peptide elimination. Showing concurrent measurements of WJ-MSC-SE33 levels would help clarify this observation.
- In the Discussion (page 12, line 369), the authors suggest that “rapid clearance of MSCs from the lungs likely explains the shorter half-life of SE-33 observed in pulmonary tissues.” Please clarify whether the clearance characteristics of WJ-MSCs align with those reported for human MSCs from other sources.SE-33 is intended for antimicrobial therapy. Please explain how the dosage criteria (for example, MIC) were established and whether these values were derived from in vitro or in vivo studies.
- The authors mention applying linear regression for concentration-time profiles. Typically, such profiles are evaluated using non-compartment analysis or compartment analysis. Please clarify the justification for applying linear regression in this context.
- Please clarify the meaning of distribution coefficient and tissue availability index. What is it for tissue availability index. It is not a common term for pharmacokinetics.
- Please specify whether the pharmacokinetic parameters were derived using non-compartmental or compartmental analysis. More detail on the methodology will help readers interpret the results. The units used for concentration, pharmacokinetic parameters, and graphical presentation appear inconsistent. Please adopt a uniform system of units throughout to prevent confusion.
Author Response
The authors describe the pharmacokinetic properties and tissue distribution of a novel therapeutic—Wharton’s jelly–derived mesenchymal stromal/stem cells (MSCs) expressing the antimicrobial peptide SE-33 (WJ-MSC-SE33). Mesenchymal stromal/stem cells have recently garnered interest as both therapeutic agents and delivery platforms. In this study, the authors utilized Wharton’s Jelly–derived MSCs to deliver the antibacterial peptide SE-33, presenting plasma concentration and tissue distribution data following intravenous administration. While this approach is promising, there are several major concerns that should be addressed:
Comment 1: Please clarify why sample collection was performed at 4 hours post–IV administration and how pharmacokinetic parameters were calculated when including the 0-hour time point.
Response 1: Thank you for pointing this out. Therefore, we have clarified the determination of the 0-hour (baseline) values in Materials and Methods section (subsection 4.5, lines 546-550).
Comment 2: Because the MSCs themselves serve as the delivery platform for SE-33, it would be beneficial to present the pharmacokinetic properties of the MSCs to elucidate the release profile of SE-33. In the Discussion (page 12, line 374), the authors attribute the lowest concentration of SE-33 in the spleen to either reduced accumulation of WJ-MSC-SE33 or rapid peptide elimination. Showing concurrent measurements of WJ-MSC-SE33 levels would help clarify this observation.
Response 2: Thank you for this valuable suggestion. We agree with the reviewer that parallel assessment of the biodistribution of WJ-MSC-SE33 would provide deeper insight into the peptide release kinetics and tissue-specific accumulation. However, we were not able to perform this analysis within the current study. We have emphasized the importance of conducting such analyses in future work in Conclusion section. Specifically, we plan to assess the biodistribution of labeled WJ-MSC-SE33 using fluorescent techniques to better correlate MSC distribution with SE-33 pharmacokinetics.
Comment 3: In the Discussion (page 12, line 369), the authors suggest that “rapid clearance of MSCs from the lungs likely explains the shorter half-life of SE-33 observed in pulmonary tissues.” Please clarify whether the clearance characteristics of WJ-MSCs align with those reported for human MSCs from other sources.SE-33 is intended for antimicrobial therapy. Please explain how the dosage criteria (for example, MIC) were established and whether these values were derived from in vitro or in vivo studies.
Response 3: Thank you for these comments. We have, accordingly, revised the Discussion section to elaborate on the known clearance characteristics of human WJ-MSCs. We have cited relevant literature to support the statement regarding rapid pulmonary clearance of WJ-MSCs (lines 398-417). Additionally, we have expanded the text to clarify the selection of administered doses of WJ-MSC-SE33 in Materials and Methods section (lines 533-543) and the previous determination of the SE-33 peptide MIC in Discussion section (lines 418-435).
Comment 4: The authors mention applying linear regression for concentration-time profiles. Typically, such profiles are evaluated using non-compartment analysis or compartment analysis. Please clarify the justification for applying linear regression in this context.
Response 4: Agree. We have, accordingly, revised Results and Discussion sections to explain our rationale for using linear regression in addition to non-compartmental analysis (lines 236-242; lines 371-392).
Comment 5: Please clarify the meaning of distribution coefficient and tissue availability index. What is it for tissue availability index. It is not a common term for pharmacokinetics.
Response 5: Thank you for highlighting this. We agree with the reviewer that the term “tissue availability index” was inaccurately used and could cause confusion. We have corrected the terminology throughout the Results and Materials and Methods sections, replacing “tissue availability index” with the more appropriate “tissue penetration factor” to describe the relative accumulation of SE-33 in different organs.
Comment 6: Please specify whether the pharmacokinetic parameters were derived using non-compartmental or compartmental analysis. More detail on the methodology will help readers interpret the results. The units used for concentration, pharmacokinetic parameters, and graphical presentation appear inconsistent. Please adopt a uniform system of units throughout to prevent confusion.
Response 6: Thank you for pointing this out. We have clarified in the Materials and Methods section that all pharmacokinetic parameters were derived using a non-compartmental analysis. We have also added more detailed descriptions of the methodology used for parameter calculation (lines 599-624). Additionally, we have revised the entire manuscript to ensure consistency in the units used for concentration, time (hours), and pharmacokinetic parameters, including in all figures and tables. These changes were made to enhance clarity and prevent misinterpretation of the data.
Reviewer 2 Report
Comments and Suggestions for Authors
The manuscript presents highly promising preclinical findings, but revisions are necessary to strengthen methodological rigor and data interpretation.
Results:
You applied linear regression analysis, but this method is not described in the methodology. Why was nonlinear compartmental analysis not used? Since you have the data, presenting it in a different format (e.g., nonlinear compartmental modelling) may be more appropriate. If you have a strong reason for using linear regression, please provide supporting evidence. Otherwise, the limitations of this approach should be explicitly discussed in the discussion section.
The study does not differentiate between free and bound SE-33 peptide. The high concentration in the liver may represent bound peptide, which may not be bioactive. Please clarify in the methodology and results what is being measured—is it total peptide, free peptide, or a mix of both?
The X-axis title in Figure 3 should be ‘Time (h)’ instead of ‘Time points’.
Methodology:
Assay Validation (Section 4.10): Has the assay method been validated? Is the assay stability-indicating? Please provide validation data (e.g., accuracy, precision, linearity, limit of detection).
Line 482: Specify which type of tube was used for heparinised blood collection (e.g., EDTA vs. lithium heparin).
Line 538: Specify the exact time points at which measurements were taken.
Supplements: The supplementary figures provide only control peaks but lack peptide peaks. Please provide chromatograms for SE-33 in each matrix (e.g., serum, liver tissue, spleen, lung) to confirm specificity and separation from endogenous components.
Comments on the Quality of English LanguageThe English could be improved to more clearly express the research.
Author Response
The manuscript presents highly promising preclinical findings, but revisions are necessary to strengthen methodological rigor and data interpretation.
Results:
Comment 1: You applied linear regression analysis, but this method is not described in the methodology. Why was nonlinear compartmental analysis not used? Since you have the data, presenting it in a different format (e.g., nonlinear compartmental modelling) may be more appropriate. If you have a strong reason for using linear regression, please provide supporting evidence. Otherwise, the limitations of this approach should be explicitly discussed in the discussion section.
Response 1: Thank you for pointing this out. We agree with this comment. Therefore, we have added a description of the linear regression analysis used in the pharmacokinetic evaluation to the Materials and Methods section (lines 624-632). We also clarified that linear regression was applied as a supplementary approach to the primary non-compartmental analysis, in order to evaluate the linearity and dose-dependence of tissue-specific SE-33 accumulation (Results, subsection 2.2). Furthermore, we have revised the Discussion section to justify the use of this method, referencing relevant literature, and to emphasize its methodological limitations in the context of complex peptide distribution and cell-based delivery systems (lines 371-392).
Comment 2: The study does not differentiate between free and bound SE-33 peptide. The high concentration in the liver may represent bound peptide, which may not be bioactive. Please clarify in the methodology and results what is being measured—is it total peptide, free peptide, or a mix of both?
Response 2: Thank you for this important remark. We agree with the reviewer and have clarified in the Materials and Methods section (line 592) that our quantification protocol measures the free fraction of the SE-33 peptide. Since tissue homogenates and serum samples were prepared without chemical or mechanical methods that would liberate protein-bound peptide, thus reflecting the unbound, and potentially bioactive, portion of SE-33.
Comment 3: The X-axis title in Figure 3 should be ‘Time (h)’ instead of ‘Time points’.
Response 3: Agree. We have, accordingly, made suggested corrections.
Methodology:
Comment 4: Assay Validation (Section 4.10): Has the assay method been validated? Is the assay stability-indicating? Please provide validation data (e.g., accuracy, precision, linearity, limit of detection).
Response 4: Thank you for pointing this out. We have revised the Materials and Methods section (subsection 4.10.4, lines 592-598) to include additional validation parameters of the assay.
Comment 5: Line 482: Specify which type of tube was used for heparinised blood collection (e.g., EDTA vs. lithium heparin).
Response 5: Thank you for noticing this. In fact, since we collected serum for peptide analysis, blood samples were obtained in tubes without anticoagulant. We have clarified this in the Materials and Methods to avoid any ambiguity (lines 550).
Comment 6: Line 538: Specify the exact time points at which measurements were taken.
Response 6: Agree. We have, accordingly, updated the Materials and Methods section to provide the precise time points used for sample collection (lines 606-608).
Supplements:
Comment 7: The supplementary figures provide only control peaks but lack peptide peaks. Please provide chromatograms for SE-33 in each matrix (e.g., serum, liver tissue, spleen, lung) to confirm specificity and separation from endogenous components.
Response 7: Thank you for this suggestion. We have updated the Supplementary Materials to include representative chromatograms for SE-33 in all biological matrices analyzed (serum, liver, lungs, and spleen). Each chromatogram now includes an arrow indicating the retention time of the SE-33 peptide to clearly demonstrate specificity and separation from endogenous compounds. Figure 1 has also been revised to include these annotations.
Response to Comments on the Quality of English Language
Point 1: The English could be improved to more clearly express the research.
Response 1: Thank you for this remark. We have thoroughly revised the manuscript to improve clarity and grammar. This included correcting inappropriate terminology, improving sentence structure, and ensuring consistency in technical language throughout the text. Our goal was to enhance readability and meet the linguistic standards expected for publication.
Round 2
Reviewer 1 Report
Comments and Suggestions for Authors
I appreciate the authors’ efforts in revising the manuscript and addressing the reviewers’ comments. The revisions have improved the clarity and quality of the work. However, a few responses would benefit from further clarification to enhance the scientific rigor and transparency of the study. With these additional clarifications, the manuscript will be a valuable contribution to the scientific community. Please find my specific comments below:
Comment 1: Please clarify why sample collection was performed at 4 hours post–IV administration and how pharmacokinetic parameters were calculated when including the 0-hour time point.
Response 1: Thank you for pointing this out. Therefore, we have clarified the determination of the 0-hour (baseline) values in Materials and Methods section (subsection 4.5, lines 546-550).
Follow up-comment: Could the authors clarify whether SE-33 was detectable at earlier time points following IV administration, such as 15 minutes, 30 minutes, or 1 hour? For small molecules, early sampling is critical because the first measurable concentration after IV dosing may reflect Cmax. Collecting early time points is essential to accurately define the concentration–time profile and calculate pharmacokinetic parameters.
Comment 5: Please clarify the meaning of distribution coefficient and tissue availability index. What is it for tissue availability index. It is not a common term for pharmacokinetics.
Response 5: Thank you for highlighting this. We agree with the reviewer that the term “tissue availability index” was inaccurately used and could cause confusion. We have corrected the terminology throughout the Results and Materials and Methods sections, replacing “tissue availability index” with the more appropriate “tissue penetration factor” to describe the relative accumulation of SE-33 in different organs.
Follow up-comment: Could the authors clarify the rationale for using “tissue penetration factor” in the context of their equation? Typically, this term refers to the tissue-to-plasma AUC ratio. It would be helpful to explicitly state how this parameter was calculated and whether it aligns with conventional definitions in pharmacokinetics.
Author Response
Comment 1: Please clarify why sample collection was performed at 4 hours post–IV administration and how pharmacokinetic parameters were calculated when including the 0-hour time point.
Response 1: Thank you for pointing this out. Therefore, we have clarified the determination of the 0-hour (baseline) values in Materials and Methods section (subsection 4.5, lines 546-550).
Follow up-comment: Could the authors clarify whether SE-33 was detectable at earlier time points following IV administration, such as 15 minutes, 30 minutes, or 1 hour? For small molecules, early sampling is critical because the first measurable concentration after IV dosing may reflect Cmax. Collecting early time points is essential to accurately define the concentration–time profile and calculate pharmacokinetic parameters.
Response 1: Thank you for this valuable comment. In our study, we addressed this issue by conducting a preliminary experiment in which samples were collected at earlier time points (1 and 2 h post-injection). These data revealed that SE-33 peptide concentrations remained below the detection limit during this period. Therefore, time points earlier than 4 hours were excluded from the main pharmacokinetic study. To clarify this rationale, we have now explicitly described this preliminary finding in the Materials and Methods section (Lines 546-549). Additionally, representative chromatograms indicating the absence of detectable SE-33 levels have been included in the Supplementary Materials (Figure S6).
Comment 5: Please clarify the meaning of distribution coefficient and tissue availability index. What is it for tissue availability index. It is not a common term for pharmacokinetics.
Response 5: Thank you for highlighting this. We agree with the reviewer that the term “tissue availability index” was inaccurately used and could cause confusion. We have corrected the terminology throughout the Results and Materials and Methods sections, replacing “tissue availability index” with the more appropriate “tissue penetration factor” to describe the relative accumulation of SE-33 in different organs.
Follow up-comment: Could the authors clarify the rationale for using “tissue penetration factor” in the context of their equation? Typically, this term refers to the tissue-to-plasma AUC ratio. It would be helpful to explicitly state how this parameter was calculated and whether it aligns with conventional definitions in pharmacokinetics.
Response 5: Thank you for this important observation. We agree with your suggestion. Therefore, we revised the corresponding Materials and Methods section (now it is subsection 4.12, Lines 621-631) to include the rationale for using this parameter and its standard definition in the context of antimicrobial pharmacokinetics. We also provided supporting references.

Reviewer 2 Report
Comments and Suggestions for Authors
It can be published in its current format.
Author Response
Comment: It can be published in its current format. Response: Dear Reviewer, thank you for your time and constructive feedback throughout the review process. We are grateful for your positive assessment.